# LATENT DIFFUSION PLANNING FOR IMITATION LEARNING

## ABSTRACT

Recent progress in robotic imitation learning has been enabled by policy archi­tectures that scale to complex visuomotor tasks, multimodal distributions, and large datasets. However, these methods rely on supervised learning of actions from expert demonstrations, which can be challenging to scale. We propose Latent Diffusion Planning, which forecasts future states as well as actions via dif­fusion. This objective can scalably leverage heterogeneous data sources and pro­vides a denser supervision signal for learning. To plan over images, we learn a compact latent space through a variational autoencoder. We then train a planner to forecast future latent states, and an inverse dynamics model to extract actions from the plans. As planning is separated from action prediction, LDP can leverage sub­optimal or action-free data to improve performance in low demonstration regimes. On simulated visual robotic manipulation tasks, LDP outperforms state-of-the-art imitation learning approaches as they cannot leverage such additional data.[1]

## 1 INTRODUCTION

Combining large-scale expert datasets and powerful imitation learning policies has been a promising direction for robot learning. Recent methods using transformer backbones or diffusion heads (Octo Model Team et al., 2024; Kim et al., 2024; Zhao et al., 2024; Chi et al., 2023) have capitalized on new robotics datasets pooled together from many institutions (Khazatsky et al., 2024; Open X-Embodiment Collaboration et al., 2023), showing potential for learning generalizable robot policies. However, this recipe is fundamentally limited by data, as robotics demonstration data is limited and expensive to collect. While it is often easier to collect in-domain data that is suboptimal or action-free, these methods are not designed to use such data, as they rely on directly modeling optimal actions.

Prior work in reinforcement learning has explored using heterogeneous data sources. Approaches that can be scaled to the imitation learning setting include conditioning the policy on optimality (Chen et al., 2021), and relabeling action-free trajectories using an inverse model (Baker et al., 2022). However, many of these approaches have not been shown to be competitive with state-of-the-art robotic imitation learning (Mirchandani et al., 2024). Recent work in robotics has leveraged heterogeneous data for pretraining via representation learning (Radosavovic et al., 2023; Wu et al., 2023b; Cui et al., 2024). However, only using the data for representation learning is limited, as it does not necessarily improve the planning capabilities of the method. In this work, we investi­gate how a simple planning-based method can leverage heterogeneous data in a principled way be decoupling forecasting future states from extracting actions.

We propose Latent Diffusion Planning (LDP), which learns a planner that can be trained on data does not require actions; and an inverse dynamics model that can be trained on data that may be sub­optimal. While prior planning-based works (Du et al., 2023a; Black et al., 2023) improve high-level decision making by producing subgoals, we focus on forecasting a dense trajectory of latent states as an alternative method for imitation learning. As diffusion objectives proved to be effective for imitation learning (Chi et al., 2023), we use diffusion for both forecasting state and actions, which enables competitive performance. LDP plans across latent image embeddings, scaling up gracefully

---

[1]We include visualizations of plans and rollouts in `https://sites.google.com/view/latent-diffusion-planning/home`

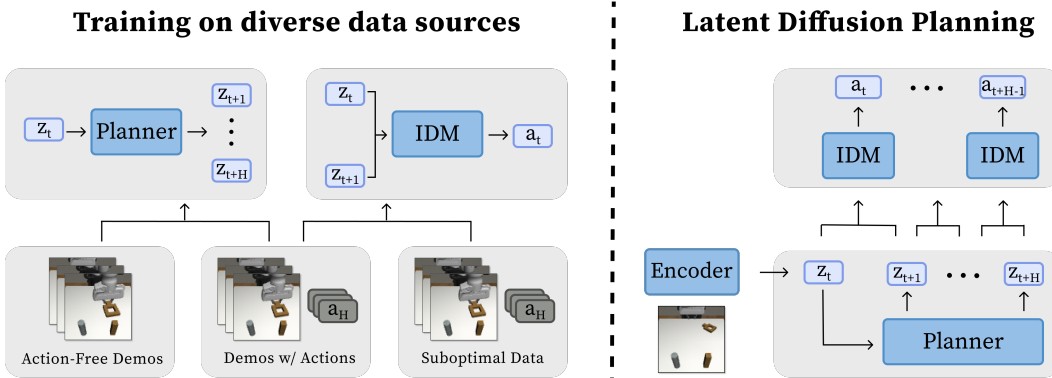

Figure 1: Latent Diffusion Planning. *Left*: LDP separates the control problem into forecasting future states with a diffusion-based planner, and extracting actions with a diffusion-based inverse dynamics model (IDM). This design enables training on heterogeneous sources of data, including suboptimal data and action-free data. *Right*: Unlike action imitation methods such as diffusion policy, LDP is based on forecasting a dense temporal sequence of latent states as well as actions. Using powerful diffusion models for both of these objectives enables LDP to have competitive performance to state-of-the-art imitation learning. Further, unlike prior work on forecasting subgoals, LDP predicts a dense temporal sequence of latent states, which enables scalable closed-loop planning.

to vision-based domains without the computational complexities of video generation. First, it trains a variational autoencoder with an image reconstruction loss, producing compressed latent embeddings. Then, it learns an imitation learning policy through two components: (1) a planner, which consumes demonstration state sequences, which may be action-free, and (2) an inverse dynamics model, trained on in-domain, possibly suboptimal, environment interactions. To maximally capture expressivity, both the planner and inverse dynamics models are implemented as diffusion models. Furthermore, our method is closed-loop and reactive, as planning over latent space is much faster than generating visually and physically consistent video frames.

In summary, our main contributions are threefold:

- We propose a novel imitation learning algorithm, Latent Diffusion Planning, a simple, diffusion planning-based method comprised of a learned visual encoder, latent planner, and an inverse dynamics model.

- We show that Latent Diffusion Planning can be trained on suboptimal or action-free data, and improves from learning on such data in the regime where demonstration data is limited. LDP can leverage such data better than prior work based on optimality conditioning or representation learning.

- We experimentally show that our method outperforms prior planning-based work by leveraging temporally dense predictions in a latent space, which enables closed-loop planning.

## 2 RELATED WORK

**Imitation Learning in Robotics.** One popular approach to learning robot control policies is imitation learning, where policies are learned from expert-collected demonstration datasets. This is most commonly done via behavior cloning, which reduces policy learning to a supervised learning objective of mapping states to actions. Recently, Diffusion Policy (Chi et al., 2023) and Action Chunking with Transformers (Zhao et al., 2023) have shown successful results in complex manipulation tasks using action chunking and more expressive architectures. Similarly, Behavior Transformer (Shafiullah et al., 2022) and VQ-BeT (Lee et al., 2024) have focused on improving the ability of policies to capture multimodal behaviors. In this work, we focus on forecasting a sequence of future states instead of actions, and use diffusion to capture multimodal trajectories.

**Learning from Unlabelled Suboptimal and Action-Free Data.** Learning from suboptimal data has long been a goal of many robot learning methods, including reinforcement learning. A typical approach is offline reinforcement learning, which considers solving a Markov decision process from an offline dataset of states, actions, and reward (Levine et al., 2020; Kumar et al., 2020; Kostrikov et al., 2021; Hansen-Estruch et al., 2023; Yu et al., 2022). Particularly relevant are the approaches that use supervised learning conditioned on rewards (Schmidhuber, 2019; Kumar et al., 2019a; Chen et al., 2021). In this work, we want to leverage suboptimal, reward-free data, such as play data or failed trajectories. In addition, we would like to avoid the additional complexity of annotating the data with rewards or training a value function which the offline RL methods rely on.

Autonomous imitation learning methods seek to self-bootstrap from a pretrained imitative policy. Typically, these methods assume learning from online, autonomous rollouts and reward labels from trained classifiers or vision-language models (Konstantinos Bousmalis* & Heess, 2023; Zhou et al., 2024b; Mirchandani et al., 2024). Unlike these works, we assume access to a static, offline dataset, and we do not label the dataset with pseudo-rewards.

Several works have also addressed learning from action-free data, such as using inverse models (Torabi et al., 2018; Baker et al., 2022), latent action models (Edwards et al., 2019; Schmeckpeper et al., 2020; Bruce et al., 2024), or representation learning (Radosavovic et al., 2023; Wu et al., 2023b; Cui et al., 2024). In this work we focus on a simple recipe for robotic imitation learning that is naturally able to leverage action free data through state forecasting.

**Diffusion and Image Prediction in Robot Learning.** Diffusion models, due to their expressivity and training and sampling stability, have been applied to robot learning tasks. Diffusion has been used in offline reinforcement learning (Hansen-Estruch et al., 2023) and imitation learning (Chi et al., 2023). Diffuser (Janner et al., 2022) learns a denoising diffusion model on trajectories, including both states and actions, in a model-based reinforcement learning setting. Decision Diffuser (Ajay et al., 2023) extends Diffuser by showing compositionality over skills, rewards, and constraints, and instead diffuses over states and uses an inverse dynamics model to extract actions from the plan. Due to the complexity of modeling image trajectories, Diffuser and Decision Diffuser restrict their applications to low-dimensional states.

To scale up to diffusing over higher-dimensional plans, UniPi (Du et al., 2023a; Ko et al., 2023) adapts video models for planning. Unlike works that rely on foundation models and video models for planning (Du et al., 2023b; Yang et al., 2024; Zhou et al., 2024a), our method avoids computational and modeling complexities of generative video modeling by planning over latent embeddings instead.

Previous works have used world models to plan over images in a compact latent space (Hansen et al., 2024; Hafner et al., 2019; 2020). In contrast with these works, we focus on single task imitation instead of reinforcement learning.

Many prior works argued that state forecasting objectives are uniquely suitable for robotics to improve planning quality with trajectory optimization or reinforcement learning Finn & Levine (2017); Yang et al. (2023), by using the model directly to plan future states Du et al. (2023b;a), as well as representation learning (Wu et al., 2023a; Radosavovic et al., 2023). We follow this line of work by proposing a planning-based method competitive to state-of-the-art robotic imitation learning that can leverage heterogeneous data sources.

## 3 BACKGROUND

**Diffusion Models** Diffusion models are likelihood-based generative models that learn an iterative denoising process from a Gaussian prior to a data distribution (Sohl-Dickstein et al., 2015; Ho et al., 2020; Song et al., 2020). Denoising Diffusion Probabilistic Models (DDPMs) (Ho et al., 2020) optimizes a variational lower bound on data likelihood, derived in a similar way to variational autoencoders (Kingma & Welling, 2014; Rezende et al., 2014). DDPMs are trained to reverse a single noising step, formally:

$$\mathcal{L}_{\text{DDPM}}(\phi, \mathbf{z}) = \mathbb{E}_{t,\epsilon}[||\epsilon_\phi(\alpha_t \mathbf{z} + \sigma_t \epsilon) - \epsilon||^2] \tag{1}$$

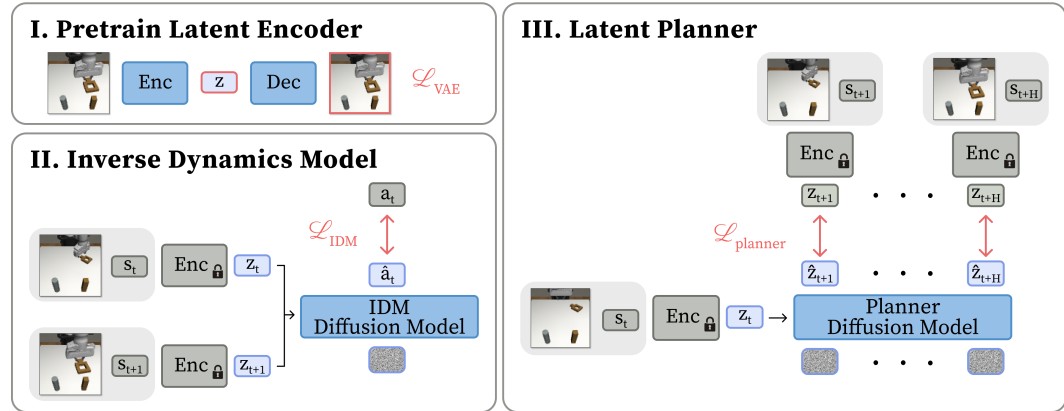

Figure 2: The architecture for Latent Diffusion Planning. *I.* We train a variational autoencoder on in-domain data to compress images into latents $z$. This allows for scalable closed-loop planning in the latent space. *II.* We train a inverse dynamics model (IDM) with a diffusion objective to directly extract the actions that will be used for control from pairs of latent states. *III.* We train a powerful latent diffusion model to forecast a chunk of future latent states. The planner and the IDM are used together to produce an action chunk, similar to Chi et al. (2023). By leveraging multi-step prediction and powerful diffusion models based on Chi et al. (2023), we can construct a method competitive to state-of-the-art imitation learning methods.

where $\mathbf{z}$ is a data sample, $\alpha_t, \sigma_t$ are noise schedule values indexed by timestep $t \in 1, 2, ..., T$, $\epsilon$ is randomly sampled Gaussian noise, and $\phi$ are learned parameters.

To reverse the diffusion process, the model iteratively denoises a sample drawn from the known prior $z_T \sim N(0, I)$. For example, DDPM samples the chain $z_T, ..., z_0$ according to:

$$p_\phi(\mathbf{z}_{t-1} \,|\, \mathbf{z}_t) = N(\mathbf{z}_{t-1} \,|\, \epsilon_\phi(\mathbf{z}_t, t), \sigma_t^2 I) \tag{2}$$

Diffusion models may also be conditioned on additional context $\mathbf{c}$. For example, text-to-image generative models are conditioned on text, Diffusion Policy is conditioned on visual observations, and Decision Diffuser can be conditioned on reward, skills, and constraints.

Recent generative models have used Latent Diffusion Models, which trains a diffusion model in a learned, compressed latent space (Rombach et al., 2022; Peebles & Xie, 2023; Blattmann et al., 2023) to improve computational and memory efficiency. The latent space is typically learned via an autoencoder, with encoder $\mathcal{E}$ and decoder $\mathcal{D}$ trained to reconstruct $x \approx \hat{x} = \mathcal{D}(\mathcal{E}(x))$. Instead of diffusing over $x$, the diffusion model is trained on diffusing over $z = \mathcal{E}(x)$.

**Imitation Learning** In the imitation learning framework, we assume access to a dataset of expert demonstrations, $\mathcal{D} \triangleq \{(s_0, x_0, a_0), .., (s_T, x_T, a_T)\}$, generated by $\pi_E$, an expert policy. $s_i, x_i, a_i$ correspond to the state, image, and action at timestep $i$ respectively. The imitation learning objective is to extract a policy $\hat{\pi}(a|s, x)$ that most closely imitates $\pi_E$. In robotics, this is typically approached through behavior cloning, which learns the mapping between states and actions directly via supervised learning. We consider single-task imitation, where the dataset corresponds to a single task.

Datasets of expert demonstrations often do not provide sufficient state distribution coverage to effectively solve a given task with imitation learning. However, there often exists additional data in the form of action-free or suboptimal data, which may consist of failed policy rollouts, play data, or miscellaneous environment interactions. Unfortunately, behavior cloning assumes access to data annotated with optimal actions, so such additional data cannot be easily incorporated into training.

# 4 LATENT DIFFUSION PLANNING

Latent Diffusion Planning consists of three stages, as shown in fig. 2: (1) Training an image encoder via an image reconstruction loss, (2) learning an inverse dynamics model to extract actions $a_t$ from pairs of latent states $z_t$, $z_{t+1}$, and (3) learning a planner to forecast future latents $z_t$.

---

**Algorithm 1** Inference with Latent Diffusion Planning

---

1: **Input:** Encoder $\mathcal{E}$, Planner $\epsilon_\psi$, IDM $\epsilon_\xi$, Planner Diffusion Timesteps $T_p$, IDM Diffusion Timesteps $T_{\text{IDM}}$, Planning Horizon $H_p$, Action Horizon $H_a$

2: Observe initial state $s_0$ and image $x_0$; $k = 0$
3: **while** not done **do**
4:    $\mathbf{z}_k \leftarrow (\mathcal{E}(x_k), s_k)$

    // Diffuse over latent embedding plan
5:    $\hat{\mathbf{z}}_{k+1}, ..., \hat{\mathbf{z}}_{k+H_p} \sim \mathcal{N}(0, I)$
6:    **for** $t = T_p \ldots 1$ **do**
7:       $\hat{\epsilon} \leftarrow \epsilon_\psi(\hat{\mathbf{z}}_{k+1}, ..., \hat{\mathbf{z}}_{k+H_p}; \mathbf{z}_k, t)$
8:       Update $\hat{\mathbf{z}}_{k+1}, ..., \hat{\mathbf{z}}_{k+H_p}$ using DDPM update with $\hat{\epsilon}$
9:    **end for**

    // Diffuse over actions between latent embeddings
10:   **for** $i = 0 \ldots H_a - 1$ **do**
11:     $\hat{a}_{k+i} \sim \mathcal{N}(0, I)$           // Predict action for each timestep in action horizon
12:     **for** $t = T_{\text{IDM}} \ldots 1$ **do**
13:       $\hat{\epsilon} \leftarrow \epsilon_\xi(\hat{a}_{k+i}; \hat{\mathbf{z}}_{k+i}, \hat{\mathbf{z}}_{k+i+1}, t)$
14:       Update $\hat{a}_{k+i}$ using DDPM update with $\hat{\epsilon}$
15:     **end for**
16:   **end for**

    // Execute actions
17:   **for** $i = 0 \ldots H_a - 1$ **do**
18:     $s_{k+i+1} \leftarrow$ env.step$(s_{k+i}, \hat{a}_{k+i})$
19:   **end for**
20:   $k \leftarrow k + H_a$
21: **end while**

---

## 4.1 LEARNING THE LATENT SPACE

We circumvent planning over high-dimensional image observations by planning over a learned latent space. Similar to prior work in planning with world models (Watter et al., 2015; Ha & Schmidhuber, 2018; Hafner et al., 2020), we learn this latent space using an image reconstruction objective. Our planner thus becomes similar to video models that forecast image frames in a learned latent space (Yan et al., 2021; Hong et al., 2022; Blattmann et al., 2023).

In practical scenarios, we may have a limited demonstration dataset, but much larger and diverse suboptimal or action-free datasets. In this phase of learning, we can make use of the visual information in such datasets for training a more robust latent encoder.

In this work, we train a variational autoencoder (Kingma & Welling, 2014; Rezende et al., 2014) to obtain a latent encoder $\mathcal{E}$ and decoder $\mathcal{D}$. Specifically, we optimize the $\beta$-VAE (Higgins et al., 2017) objective:

$$\mathcal{L}_{\text{VAE}}(\theta, \phi; \mathbf{x}, \mathbf{z}, \beta) = \mathbb{E}_{q_\phi(\mathbf{z} \mid \mathbf{x})}[\log p_\theta(\mathbf{x} \mid \mathbf{z})] - \beta \mathcal{D}_{\text{KL}}(q_\phi(\mathbf{z} \mid \mathbf{x}) || p(\mathbf{z})) \tag{3}$$

where $\mathbf{x}$ is our original image, $\mathbf{z}$ is our learned latent representation of the image, $\theta$ are the parameters for our decoder, $\phi$ are the parameters for our encoder, and $\beta$ is the weight for the KL regularization term.

### 4.2 PLANNER AND INVERSE DYNAMICS MODEL

Our policy consists of two separate modules: (1) a planner over latent embeddings, and (2) an inverse dynamics model. The planner and IDM both optimize the DDPM objective.

The planner is conditioned on the current latent embedding, which consists of the concatenated latent image embedding and robot proprioception, and diffuses over a horizon of future embeddings. We use Diffusion Policy's Conditional U-Net architecture. Concretely, we optimize the following objective:

$$\mathcal{L}_{\text{planner}}(\psi, \mathbf{z}) = \mathbb{E}_{t,\epsilon}[||\epsilon_\psi(\hat{\mathbf{z}}_{k+1}, ..., \hat{\mathbf{z}}_{k+H}; \mathbf{z}_k, t) - \epsilon||^2] \quad (4)$$

where $\mathbf{z}_k$ is the latent embedding at timestep $k$ of the trajectory; $\hat{\mathbf{z}}_{k+1}, ..., \hat{\mathbf{z}}_{k+H}$ is the noised latent embedding sequence, with corresponding noise $\epsilon$; $H$ is the maximum horizon of the forecasted latent plan; $t$ is the diffusion noise timestep; and $\psi$ are the parameters of the planner diffusion model.

Our inverse dynamics model is trained to reconstruct the action between a pair of states, conditioned on their associated latent embeddings.

$$\mathcal{L}_{\text{IDM}}(\xi, \mathbf{z}) = \mathbb{E}_{t,\epsilon}[||\epsilon_\xi(\hat{a}_k; \mathbf{z}_k, \mathbf{z}_{k+1}, t) - \epsilon||^2] \quad (5)$$

where $\mathbf{z}_k$ is the latent embedding at timestep $k$ of the trajectory; $\hat{a}_k$ is the noised action, with corresponding noise $\epsilon$; $t$ is the diffusion noise timestep; and $\xi$ are the parameters of the inverse dynamics diffusion model.

During inference, the planner forecasts a future horizon of states. Like Diffusion Policy, we employ receding-horizon control (Mayne & Michalska, 1988), and execute for a shorter horizon than the full forecasted horizon. We use the inverse dynamics model to extract actions from latent embedding pairs produced by the planner. We use DDPM sampling for both the planner and inverse dynamics models.

## 5 EXPERIMENTS

We seek to answer the following questions:

- Is Latent Diffusion Planning an simple and effective imitation learning algorithm, compared to state-of-the-art imitation learning algorithms or methods that may leverage suboptimal data?
- Does our method leverage action-free data for improved planning?
- Does Latent Diffusion Planning enable us to effectively utilize and scale favorably with suboptimal data?

### 5.1 EXPERIMENTAL SETUP

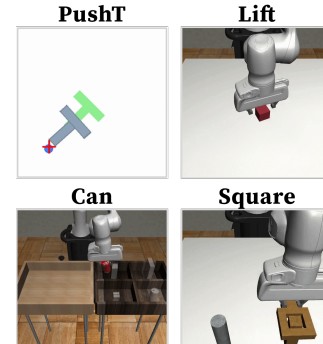

**PushT    Lift**

**Can    Square**

**Tasks** We focus our experiments on 4 image-based imitation learning tasks: (1) PushT, (2) Robomimic Lift, (3) Robomimic Can, and (4) Robomimic Square. PushT, adapted from IBC and Diffusion Policy (Florence et al., 2021; Chi et al., 2023), involves pushing a block to a target position with 2D end-effector control. Robomimic (Mandlekar et al., 2021) is a robotic manipulation and imitation benchmark.

**Dataset** To demonstrate the effectiveness of Latent Diffusion Planning, we assume a low-demonstration data regime. For PushT, Can, and Square, we filter 100 demonstrations out of the 200 total, and for Lift, we filter 3 demonstrations out of the 200 total. To further emphasize the importance of suboptimal data, these demonstrations cover a limited state space of the entire environment. For PushT, we filter demonstrations such that the agent never reaches the right third of the 2D state space. For Robomimic tasks, we filter demonstrations based on the initialization of the object of interest, such that the initialization does not cover the entire distribution of object initializations during evaluations.

Table 1: **Leveraging Suboptimal Data.** Latent Diffusion Planning, outperforms prior imitation learning works as it better utilizes suboptimal data via VAE representation learning and training the inverse dynamics model.

| Method | PushT | Lift | Can | Square |
|---|---|---|---|---|
| DP | 0.29 ± 0.002 | 0.36 ± 0.000 | 0.43 ± 0.010 | 0.37 ± 0.010 |
| RC-DP | **0.58** ± 0.000 | 0.38 ± 0.08 | 0.43 ± 0.030 | **0.50** ± 0.060 |
| DP+Repr | 0.19 ± 0.005 | 0.52 ± 0.020 | **0.56** ± 0.020 | 0.42 ± 0.040 |
| UniPi-OL | 0.18 ± 0.006 | 0.47 ± 0.050 | 0.10 ± 0.020 | 0.15 ± 0.070 |
| UniPi-CL | 0.51 ± 0.032 | 0.14 ± 0.020 | 0.34 ± 0.020 | 0.11 ± 0.010 |
| LDP + Subopt (ours) | 0.20 ± 0.013 | **0.83** ± 0.030 | **0.58** ± 0.020 | **0.47** ± 0.010 |

Our suboptimal data consists of failed trajectories from an under-trained behavior cloning agent. For simplicity, we assume an observation horizon of 1 and a single-view image input for all tasks. We use the third-person camera for Robomimic.

**Baselines**

- **Diffusion Policy** (**DP**) is a state-of-the-art imitation learning algorithm.

- **Reward-Conditioned Diffusion Policy** (**RC-DP**) utilizes suboptimal actions by conditioning the policy on a binary value indicating whether the action chunk comes from optimal demonstrations or not. This method is inspired by reward-conditioned approaches (Kumar et al., 2019b; Chen et al., 2021)

- **Diffusion Policy with Representation Learning** (**DP+Repr**) uses a VAE pretrained on demonstration and suboptimal data as the observation encoder. This is representative of the methods that leverage suboptimal data through representation learning.

- **Open-Loop UniPi** (**UniPi-OL**) is based off of UniPi (Du et al., 2023a), a video planner for robot manipulation. UniPi-OL generates a single video trajectory, extracts actions, and executes the actions in an open-loop fashion. We use a goal-conditioned behavior cloning agent to reach generated subgoals (Wen et al., 2024).

- **Closed-Loop UniPi** (**UniPi-CL**) is a modification that allows UniPi to perform closed-loop replanning over image chunks. Like LDP, UniPi-CL generates dense plans instead of waypoints, though in image space. We learn an inverse dynamics model to extract actions.

Table 2: **Leveraging Action-Free Data.** LDP outperforms prior imitation learning works as it better utilizes action free data via a state forecasting objective.

| Method | Lift | Can | Square |
|---|---|---|---|
| DP | 0.36 ± 0.000 | 0.43 ± 0.010 | 0.37 ± 0.010 |
| UniPi-OL + Action-Free | 0.48 ± 0.060 | 0.15 ± 0.010 | 0.21 ± 0.03 |
| LDP + Action-Free (ours) | **0.55** ± 0.030 | **0.99** ± 0.010 | **0.40** ± 0.020 |

## 5.2 IMITATION LEARNING WITH SUBOPTIMAL DATA

In table 1, we present imitation learning results. First, LDP outperforms DP, which can only utilize data with optimal actions. LDP, which uses suboptimal data for the VAE and IDM, can leverage diverse data sources outside of the demonstration dataset.

RC-DP, a conditional variant of DP that utilizes suboptimal data, achieves competitive results for PushT Square, while struggling to improve for Lift or Can. We hypothesize that for the Square task, the primitive motions of reaching or grasping the object, which are partially covered by the suboptimal dataset, provides a useful visuomotor prior for the policy. In addition, the suboptimal data from PushT may provide a useful prior in how to interact with the object. LDP outperforms RC-DP as it is able to leverage additional data directly for better action extraction, whereas RC-DP and DP+Repr only use it to learn better representations.

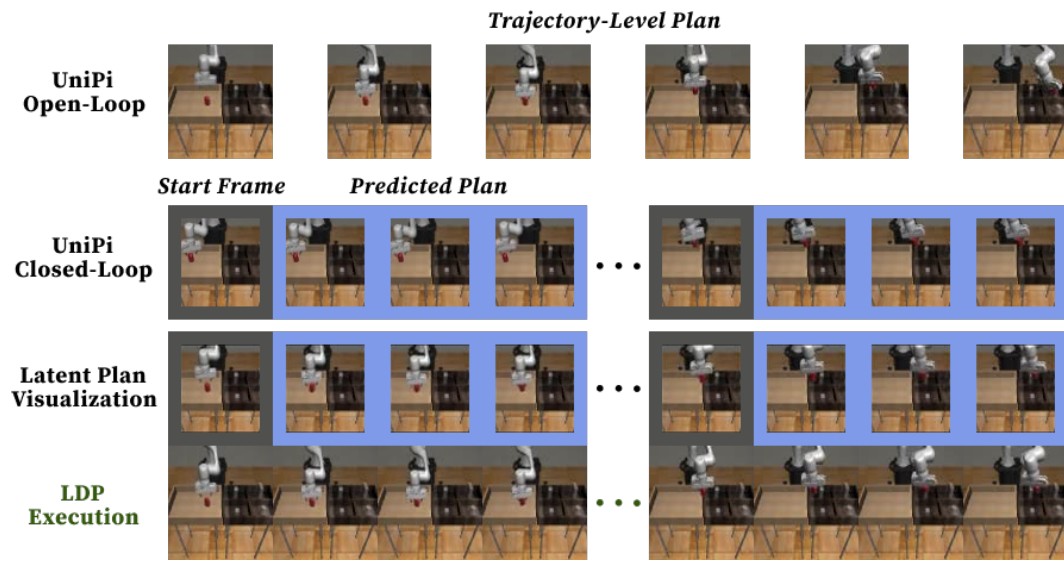

Figure 3: Visualizations of Generated Plans. UniPi-OL generates an entire trajectory for a GCBC agent to follow. UniPi-CL generates single-step image sequences. LDP generates latent plans (visualized via the VAE decoder), using an IDM to execute the plan.

In addition, we consider using a pretrained encoder to extract image features for DP (DP-Repr). We find that learning strong representations through pretraining leads to consistent improved performance for all Robomimic tasks.

Next, we compare against UniPi, which plans over image subgoals (OL) or image chunks (CL). Due to the low demonstration data regime, learning effective and accurate video policies is difficult, and LDP strongly outperforms UniPi-OL. LDP also outperforms UniPi-CL in all Robomimic environments. We hypothesize that this is due to the difficulties learning to forecast dense image chunks. For that reason, the high performance of UniPi-CL on PushT may be attributed to the simpler environment dynamics and observations.

### 5.3 IMITATION LEARNING WITH ACTION-FREE DATA

Imitation learning policies that model actions, such as DP, are unable to use action-free data, while planning-based approaches can benefit from this additional data. We train the UniPi video with additional action-free demonstrations. We find that this leads to a slight boost in performance, compared to results from table 1, but it still does not outperform LDP. We also train the LDP planner with additional action-free demonstrations. We use a VAE pretrained with demonstration and suboptimal images, and only add the action-free data for the planner, to isolate the effect of action-free data for the planner. We find that action-free demonstrations leads to a large increase in performance in Can, although surprisingly, it does not improve performance for Lift or Square. The improvement in Can may be due to the visual complexity of the scene, where action-free data may provide additional reasoning. Both the third-person Lift and Square views have minimal table or background textures, whereas the Can task observations are comparatively zoomed out, with two unique table textures and a noticeable floor texture.

### 5.4 ABLATIONS: SUBOPTIMAL DATA

Suboptimal data can be used for (1) training the VAE (any image data, whether action-labeled or action-free, suffices), and (2) training the IDM. table 3 shows the effects of using each type of data across different suboptimal dataset sizes: no suboptimal data, 1,000 suboptimal trajectories, and 2,000 suboptimal trajectories.

Table 3: **Scaling the Amount of Suboptimal Data.** We investigate the effects of using suboptimal data for pretraining the VAE encoder and for training the IDM. The size of the suboptimal dataset minimally affects performance, but using the data for both encoder and the IDM leads to stronger performance.

| Method | Lift | Can | Square |
|---|---|---|---|
| No Subopt | $0.50 \pm 0.030$ | $0.40 \pm 0.020$ | $0.25 \pm 0.030$ |
| Subopt 1k (Encoder) | $0.74 \pm 0.060$ | $\mathbf{0.63} \pm 0.010$ | $0.39 \pm 0.010$ |
| Subopt 1k (Encoder & IDM) | $\mathbf{0.83} \pm 0.030$ | $0.54 \pm 0.000$ | $\mathbf{0.47} \pm 0.030$ |
| Subopt 2k (Encoder) | $0.77 \pm 0.090$ | $0.58 \pm 0.020$ | $0.37 \pm 0.010$ |
| Subopt 2k (Encoder & IDM) | $\mathbf{0.83} \pm 0.030$ | $0.58 \pm 0.020$ | $\mathbf{0.47} \pm 0.010$ |

First, we find that not using suboptimal data drastically hurts performance. Only using suboptimal data for the VAE training dramatically improves performance, as it improves the quality of latent embeddings used for both planning and action extraction. Qualitatively and quantitatively, we find that image reconstruction on evaluation images from each environment improves with the use of suboptimal data. Visualized plans naturally look more cohesive, because the underlying latent embeddings used for planning are better structured.

Next, we find that using suboptimal data for inverse dynamics typically improves performance further. This suggests that additional environment interactions helps learn a more generalizable IDM, which can more faithfully extract actions from the diffused plan.

Scaling the amount of suboptimal data from 1,000 to 2,000 trajectories does not lead to noticeable improvements in performance. We hypothesize that the environments may be simple enough that further data does not bring extraordinary benefits. In addition, the difference between the two datasets may be minimal, and using even larger suboptimal datasets or different types of suboptimal data may lead to further improvements.

## 6    DISCUSSION

We presented Latent Diffusion Planning, a simple planning-based method for imitation learning. We show that our design using powerful diffusion models for latent state forecasting enables competitive performance with state-of-the-art imitation learning. We further show this latent state forecasting objective enables us to easily leverage heterogeneous data sources. In low-demonstration data imitation learning regime, LDP outperforms prior imitation learning work that does not leverage such additional data as effectively.

**Limitations.** One limitation of the current approach is that the latent space for planning is simply learned with a variational autoencoder and might not learn the most useful features for control. Future work will explore different representation learning objectives. Furher, our method requires diffusing over states, which incurs additional computational overhead as compared to diffusing actions. However, we expect continued improvements in hardware and inference speed will mitigate this drawback. Finally, we did not explore applying recent improvements in diffusion models (Peebles & Xie, 2023; Lipman et al., 2022), which will be important to scale to real-world applications.

**Future work.** We have validated in simulation the hypothesis that latent state forecasting can leverage heterogeneous data sources. Future work will evaluate whether this can be used to improve practical real-world applications. One direction is to use a diverse dataset of human collected data, such as with handheld data collection tools (Young et al., 2021). Another approach would be to use autonomously collected robotic data (Konstantinos Bousmalis* & Heess, 2023). As these alternative data sources are easier to collect than demonstrations, they represent a different scaling paradigm that can outperform pure behavior cloning approaches. By presenting a method that can leverage such data, we believe this work makes a step toward more performant and general robot policies.

## 7 REPRODUCIBILITY STATEMENT

For reproducibility, we will open-source the implementations for our method. Our work primarily builds upon existing work Chi et al. (2023); Du et al. (2023a); Hansen-Estruch et al. (2023); Peebles & Xie (2023), which are also publicly available. In the Appendix, we include implementation details and hyperparameters.

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

# A APPENDIX

## A.1 IMPLEMENTATION DETAILS

**Diffusion Policy** We use a Jax reimplementation of the convolutional Diffusion Policy, which we verify can reproduce reported Robomimic benchmark results. For improved performance, we process the 512-dimensional ResNet feature with an MLP [512, 256, 32] with ReLU activations and a final tanh activation.

**UniPi** We use the open-source implementation of UniPi (Ko et al., 2023). For UniPi-OL and UniPi-CL, we predict 7 future frames. During training time, for UniPi-OL, the 7 future frames are evenly sampled from a training demonstration. For UniPi-CL, the 7 future frames are the next consecutive frames.

The goal-conditioned behavior cloning agent is implemented as a goal-conditioned Diffusion Policy (Chi et al., 2023) agent, and it is trained on chunks of 16. The inverse dynamics model is based off of Hansen-Estruch et al. (2023), and shares the same architecture as the IDM used in LDP.

We train the video prediction models for 100k gradient steps with batch size 16.

**LDP** The LDP VAE is adapted from Diffusion Transformer (Peebles & Xie, 2023). The planner is based directly off of the convolutional U-Net from Diffusion Policy (Chi et al., 2023), with modifications to plan across latent embeddings instead of action chunks. The IDM is based off of Hansen-Estruch et al. (2023).

Table 4: Diffusion Policy Architecture Hyperparameters

|  | UniPi-OL GCBC | DP and LDP | LDP - Square |
|---|---|---|---|
| down_dims | [256, 512, 1024] | [256, 512, 1024] | [256, 512, 1024, 2048] |
| n_diffusion_steps | 100 | 100 | 100 |
| batch_size | 512 | 256 | 256 |
| lr | 1e-4 | 1e-4 | 1e-4 |
| n_grad_steps | 200k | 500k | 500k |

Table 5: IDM Architecture Hyperparameters

|  | UniPi-CL IDM | LDP IDM |
|---|---|---|
| n_blocks | 3 | 5 |
| n_diffusion_steps | 100 | 100 |
| batch_size | 512 | 256 |
| lr | 1e-4 | 1e-4 |
| n_grad_steps | 200k | 500k |

Table 6: VAE Architecture Hyperparameters

|  | VAE |
|---|---|
| block_out_channels | [128, 256, 256, 256, 256, 256] |
| down_block_types | [DownEncoderBlock2D] x5 |
| up_block_types | [UpDecoderBlock2D] x5 |
| latent_channels | 4 |
| PushT Latent Dim | (3, 3, 4) |
| Robomimic Latent Dim | (2, 2, 4) |
| PushT KL Beta | 1e-5 |
| Lift KL Beta | 1e-5 |
| Can KL Beta | 5e-6 |
| Square KL Beta | 5e-6 |
| n_grad_steps | 300k |

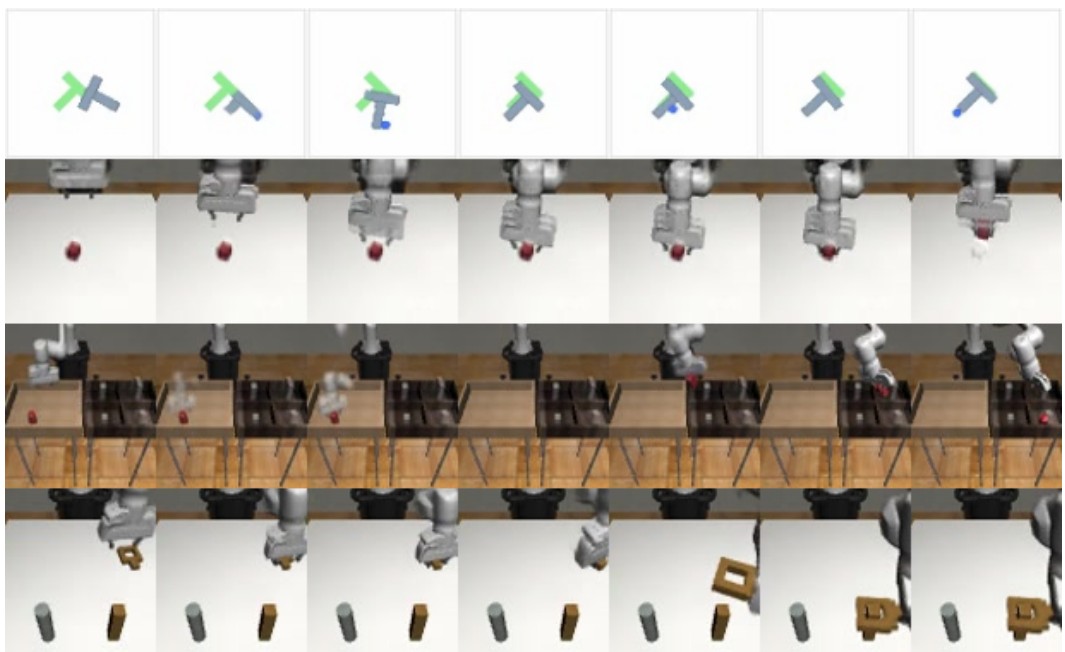

Figure 4: Visualizations of UniPi-OL plans. In PushT, there are small visual mistakes, such as a deformed T or a missing agent. Can, a more visually complex environment, also suffers from this challenge.

## A.2 SIMULATION EXPERIMENTS

In our experiments, we report results on 2 seeds, across the best performing checkpoint from last 5 saved checkpoints. PushT results are based on environment reward, and Robomimic results are reported as success rates. For UniPi, we train two goal-conditioned or inverse dynamics models and report success from the 200k checkpoint.

For UniPi-OL evaluations, we predetermine the number of steps for the GCBC to reach each image subgoal based on the demonstration lengths. For PushT, evaluation episode lengths are 200 steps; Lift is 60 steps; Can is 140 steps; and Square is 160 steps. This maximum horizon is also enforced for UniPi-CL evaluations, for consistency.

## A.3 UNIPI PLAN VISUALIZATIONS

We include visualizations of closed-loop replanning from UniPi-CL and LDP on our website: https://sites.google.com/view/latent-diffusion-planning/home

We include examples of non-cherrypicked UniPi-OL plans (trained w/o action-free data) in fig. 4.

