# OpenReview forum: "Latent Diffusion Planning for Imitation Learning"
_ICLR.cc/2025/Conference — Submitted to ICLR 2025_

### Official Review · Reviewer_UCfM · 2024-10-27

**Soundness:** 2
**Presentation:** 2
**Contribution:** 2
**Rating:** 3
**Confidence:** 4

**Summary:**

This paper proposes LDM, which learns a latent video diffusion model and an inverse dynamics model, to enable learning from action-free data as well as suboptimal data. The authors test their method via 4 imitation learning task to show the efficiency of their method.

**Strengths:**

The paper is well-written, and the background and related work section is comprehensive.

**Weaknesses:**

The major concern is novelty. Both the proposed method (LDP) and UniPi seems to (1) Learn a video planner to generate future frames; (2) learn an inverse dynamics model to extract the actions. The difference seems to exist in the selection of video planner, where UniPi uses Video Diffusion Model while this paper uses latent video diffusion model, which, from my perspective, is not novel enough to support a new paper.

Can the authors clarify what they consider to be the novel contributions of their approach compared to UniPi and other related work?

Some experiment settings remain unclear to me, see the questions below.

**Questions:**

1. Can you further explain the differences between the proposed Latent Diffusion Planning with the UniPi?

2. Can you provide some intuitions on why unipi+OL may outperform unipi+CL in 2 out of 4 tasks? Since closed-loop control is believed to be better than open-loop control.

3. In Section 5.1, the authors mentioned they filtered "3 demonstrations out of the 200 total" for the task "lift". What does this mean? How many trajectories did you use to train the video diffusion policy?

4. How many trajectories are used for other two tasks (i.e., Can, and Square)?  Can you provide the detailed number of how many trajectories are used to train each component?

---

### Official Review · Reviewer_Ny7S · 2024-10-29

**Soundness:** 2
**Presentation:** 2
**Contribution:** 3
**Rating:** 3
**Confidence:** 3

**Summary:**

This work presents Latent Diffusion Planning (LDP). LDP consists of three sub-models. A latent encoder encodes the input data into a latent space, a diffusion-based inverse dynamics model that predicts the actions for two subsequent latent embeddings, and a diffusion-based latent planner that predicts the latent states of the system for a horizon H. As opposed to traditional behavior cloning, LDP can deal with action-free and suboptimal demonstrations which is showcased in various experiments.

**Strengths:**

I find the idea and the research conducted important for the community, given that qualitative data in robotics is difficult to collect. Enhancing methods that can extract important information from any data is essential.

**Weaknesses:**

The presentation of the paper needs improvement:
- While the quality of some images is good, others need improvement (e.g. Fig. 3). A higher resolution would be useful.
- Fig 1 + Fig 2 seem to be rather similar. Does it make sense to merge them to save some space and use this space to provide more details about the method in the text?
- Fig 3 is oversized and does not provide an intuition.
- Presentations of the results are poor. The main text does not say the metrics reported in the tables and how many seeds were conducted. It is hard to judge whether the reported comparisons are convincing


Missing related works:
A quick Google search with "diffusion imitation learning" already proposed two works that have not been cited:
-Tim Pearce et al "Imitating Human Behaviour with Diffusion Models" ICLR 2023
-Moritz Reuss et al "Goal Conditioned Imitation Learning using Score-based Diffusion Policies" RSS 2023
There might be a need for more detailed literature research.

**Questions:**

-The paper claims that "Latent Diffusion Planning can be trained on suboptimal or action-free data and improves from learning on such data in the regime where demonstration data is limited". Is this claim justified given that the inverse dynamics model needs action targets to be trained?  It is unclear to me how action-free learning is done. This should be discussed in more detail in Section 4.

-Why is the performance for the PushT significantly worse than compared to other experiments in Table 1? Is there an intuition for this observation?

---

### Official Review · Reviewer_N1HK · 2024-11-01

**Soundness:** 2
**Presentation:** 2
**Contribution:** 3
**Rating:** 3
**Confidence:** 4

**Summary:**

This paper proposes Latent Diffusion Planning (LDP), an imitation learning framework that combines a learned visual encoder, a latent planner, and an inverse dynamics model within a diffusion planning approach. LDP can utilize suboptimal and action-free data, enhancing its applicability across diverse data sources. The authors validate the framework’s performance on the Robomimic benchmark, where LDP outperforms baselines by leveraging temporally dense predictions and heterogeneous data.

**Strengths:**

This work addresses the limitations of imitation learning systems that rely on high-quality, action-annotated datasets. Many real-world data sources are suboptimal or lack explicit action labels, limiting the applicability of traditional imitation learning approaches. By creating a framework that can learn effectively from suboptimal and action-free data, LDP broadens the scope of imitation learning to include a more diverse range of real-world scenarios.

**Weaknesses:**

The current experimental results and ablation studies do not fully illustrate that LDP is the most effective method to leverage suboptimal and action-free data. To clarify the method's effectiveness, the following additional experiments and details would be beneficial:

### Action-Free Data
- **Quantity of Action-Free Data:** The number of action-free data points used in the experiments is missing. Including this information would provide greater clarity on data usage.
- **Ablation on Action-Free Data Quantity:** An ablation study analyzing how varying amounts of action-free data impact the results would help understand its role in LDP's effectiveness.

### Leveraging Both Suboptimal and Action-Free Data
- **LDP with Suboptimal and Action-Free Data:** An experiment combining LDP with suboptimal and action-free data would highlight LDP's ability to leverage diverse data sources effectively.
- **IDM-Labeled Action-Free Data for DP Training:** Utilizing action-free data labeled by Inverse Dynamics Modeling (IDM) for DP training could be a potential baseline.

**Questions:**

1. This paper uses an under-trained behavior cloning agent to generate suboptimal data.
   - **a.** I’m curious about what dataset the BC agent is trained on. This could clarify how this suboptimal data may impact the learning process and model performance.
   - **b.** Why did you choose this method to generate suboptimal data? It seems unrealistic in real-world problems.
   - **c.** Some works address imitation learning from suboptimal data [1]-[5]. Comparing with those baselines could help illustrate your framework’s effectiveness.

2. Result Analysis:
   - **a.** Why does LDP perform worse on PushT? Further analysis could help understand the framework’s limitations.
   - **b.** In Table 3, the performance saturates or even drops when suboptimal data increases from 1k to 2k. Did you try 0.5k to capture the trend further? Why does using suboptimal data to train IDM hurt the performance in Can?

[1] Sasaki, Fumihiro, and Ryota Yamashina. "Behavioral cloning from noisy demonstrations." International Conference on Learning Representations. 2020.

[2] Kim, Geon-Hyeong, et al. "Demodice: Offline imitation learning with supplementary imperfect demonstrations." International Conference on Learning Representations. 2022.

[3] Yu, Lantao, et al. "Offline imitation learning with suboptimal demonstrations via relaxed distribution matching." Proceedings of the AAAI conference on artificial intelligence. Vol. 37. No. 9. 2023.

[4] Xu, Haoran, et al. "Discriminator-weighted offline imitation learning from suboptimal demonstrations." International Conference on Machine Learning. PMLR, 2022.

[5] Zhang, Wenjia, et al. "Discriminator-guided model-based offline imitation learning." Conference on Robot Learning. PMLR, 2023.

---

### Official Review · Reviewer_uJJ8 · 2024-11-02

**Soundness:** 3
**Presentation:** 3
**Contribution:** 2
**Rating:** 3
**Confidence:** 4

**Summary:**

This paper introduces Latent Diffusion Planning (LDP), a novel imitation learning approach that leverages a diffusion-based latent planner alongside a diffusion-based action prediction model. The planner is designed to learn from heterogeneous and action-free data sources, enhancing its versatility. An inverse dynamics model is then used to predict actions based on the plans. Such a hierarchical design improves the policy’s performance using limited demonstrations. The authors demonstrate the effectiveness of LDP in 4 simulation tasks

**Strengths:**

- The LDP can be trained from different data sources, which is quite important for robot learning tasks. Pretraining on larger action-free datasets for the latent planner could be a future direction.
- Using diffusion models in the latent planner is ideal for learning the multi-modality.
- The LDP achieves state-of-the-art performance compared to other imitation learning methods. Especially on the Lift task, with only 3 demonstrations, it achieves a 0.83 success rate.
- The paper is well-written with the idea of LDP explained clearly.

**Weaknesses:**

- The authors only use a VAE to pretrain a latent encoder, which could be improved by other powerful representation learning models.
- In Table 1, LDP seems to perform much better on the Lift task while performing worse on the PushT task. On the Can and Square tasks, LDP performs on par with DP+Repr. The results are not that convincing.
- The authors only conduct experiments on 4 tasks with pushing and pick-placing actions. Furthermore, the background for these tasks is relatively simple. More complex scenarios would make the latent representations hard to learn, which could be challenging for LDP.
- In the experiments, the authors take a single-view image as the input. However, in many other simulation tasks or real-world tasks, multiple cameras including an in-hand camera are employed. It would be good to see how the LDP scale with multiple cameras.
- The idea of training the latent planner on action-free data is interesting. However, the authors only evaluate their models with the datasets that provide actions. It would be nice to show if pretraining on other data sources would improve the performance. Real-world experiments would greatly enhance the quality of the paper.

**Questions:**

Overall, I like the idea of the paper, but I am concerned about the experiments part. I have the following questions:

- Why choose $\beta$-VAE? It is usually used for disentangled representation learning and the $\beta$ could influence the reconstructions.
- In the imitation learning demonstrations, the differences between two frames are very small, which means the latent from the next frame could provide little information. How does this architecture perform better than imitation learning with sub-goals?
- If the IDM takes the full predicted state latent, would the LDP perform better or worse?
- What is the length of state latent in the planner? When comparing LDP with DiffusionPolicy, do the authors use the same length of actions?
- Why does the LDP perform poorly on the PushT task?
- If using multiple cameras, the LDP might require multiple diffusion planners which could be hard to train and do inference. How do the authors expect to solve this? Further, I am wondering if the planner could perform well on the in-hand camera.
- The authors mentioned the speed drawbacks of the LDP. What is the inference time for the LDP?

---

### Official Review · Reviewer_t7eX · 2024-11-02

**Soundness:** 2
**Presentation:** 4
**Contribution:** 3
**Rating:** 5
**Confidence:** 4

**Summary:**

This paper presents Latent Diffusion Planning, a method for imitation learning in robotics that forecasts futures states and actions using a diffusion model (planning), and an inverse dynamics model to extract actions from generated plans. They implement both the planner and inverse prediction models as diffusion models. In contrast to video prediction methods for planning, this paper plans in a latent space (learned using a VAE with an image reconstruction objective) which is much faster as it does not require time-consuming frame generation.

**Strengths:**

This paper presents a novel approach for imitation learning in robotics, that breaks down decision making into a latent-space planning and inverse-dynamics prediction modules. The paper is well-written, with the key contributions listed clearly in the introduction. The authors also identified relevant literature in this line of research and highlighted their contributions in the method. The idea of inverse dynamics prediction is not novel, however this particular approach that leverages diffusion models for both components (planning and control) is novel.

**Weaknesses:**

1. The paper’s original claims supporting the method were around the ability to leverage heterogeneous datasets for imitation learning in robotics. These claims do not seem to be well-supported in the experiments. The scope of heterogeneity is another set of sub-optimal teleoperation demonstrations in the same domain (MuJoCo simulations).
2. The authors experimented with 3 simulated tasks with no experiments on a real robot, which leaves the question open as to whether this method will scale to a real robotics setup.

**Questions:**

1. What was the scope of action-less datasets that were used to train the planner? Have the authors considered training on large-scale video datasets, rather than just sim-only robot videos? Will this method be able to leverage that much heterogeneity in the datasets?
2. The main contribution of UniPi (if I understand correctly) was combinatorial generalization to unseen language instructions. Can the LDP leverage their video prediction model to handle such generalization to novel goals?
3. Can the authors report performance using a baseline BC-RNN model on the datasets they curated for their tasks?
4. How many rollouts were performed to compute success rates? Sorry if I missed, but I could not find this in the paper.
5. The tasks that this paper used are part of a suite has multiple other tasks such as coffee, tool hang, mug cleanup, etc. Why were just the 3 simulated tasks chosen for experimentation?

---

### Meta-Review · Area_Chair_m13b · 2024-12-16

**Metareview:**

All the reviewers unanimously recommended rejecting this paper concerning its clarity, insufficient experiments, experimental setup, and missing related work, while the authors did not provide a rebuttal to address these issues. Consequently, I recommend rejecting the paper.

Additionally, I would like to urge the authors to at least show appreciation for the efforts the reviewers put into helping them improve the submission.

**Additional Comments On Reviewer Discussion:**

N/A: the authors have not provided a rebuttal.

---

### Decision · Program_Chairs · 2025-01-22

Reject